# Virtual and Augmented Reality for Chronic Musculoskeletal Rehabilitation: A Systematic Review and Exploratory Meta-Analysis

**DOI:** 10.3390/bioengineering12070745

**Published:** 2025-07-08

**Authors:** Theodora Plavoukou, Pantelis Staktopoulos, Georgios Papagiannis, Dimitrios Stasinopoulos, George Georgoudis

**Affiliations:** 1Laboratory of Musculoskeletal Physiotherapy, Department of Physiotherapy, Faculty of Health and Caring Sciences, University of West Attica, Egaleo Campus, 12243 Athens, Greece; pantelisstaktopoulos@gmail.com (P.S.); dstasinopoulos@uniwa.gr (D.S.); 2Biomechanics Laboratory, Physiotherapy Department, University of the Peloponnese, 23100 Sparta, Greece; grpapagiannis@yahoo.gr; 3Department of Physiotherapy, University of West Attica, 12243 Athens, Greece; gg.physio@gmail.com

**Keywords:** virtual reality, augmented reality, exergaming, musculoskeletal disorders, rehabilitation, meta-analysis, chronic pain, physiotherapy

## Abstract

**Background:** Chronic musculoskeletal disorders (CMDs) represent a leading cause of global disability and diminished quality of life, and they are often resistant to conventional physiotherapy. Emerging technologies such as virtual reality (VR), augmented reality (AR), and exergaming are increasingly used to enhance rehabilitation outcomes, yet their comparative effectiveness remains unclear. **Objective:** To systematically evaluate the effectiveness of VR, AR, and exergaming interventions in improving pain, function, balance, and psychological outcomes among adults with CMDs. **Methods:** This systematic review and exploratory meta-analysis followed PRISMA 2020 guidelines and was prospectively registered (PROSPERO: CRD42024589007). A structured search was conducted in PubMed, Cochrane CENTRAL, Scopus, and PEDro (up to 1 May 2025). Eligible studies were randomized controlled trials (RCTs) involving adults (≥18 years) with CMDs receiving VR, AR, or exergaming-based rehabilitation. Risk of bias was assessed using the PEDro scale and the Downs and Black checklist. Where feasible, standardized mean differences (SMDs) for pain outcomes were pooled using a random-effects model. **Results:** Thirteen RCTs (*n* = 881 participants) met the inclusion criteria. Interventions spanned immersive VR, AR overlays, exergaming platforms (e.g., Kinect, Wii), and motion-tracking systems. Pain, function, and quality of life improved in most studies. An exploratory meta-analysis of eight RCTs (*n* = 610) yielded a significant pooled effect favoring VR/AR interventions for pain reduction (SMD = −1.14; 95% CI: −1.63 to −0.75; I^2^ = 0%). Exergaming showed consistent improvements in physical performance, while immersive VR was more effective for kinesiophobia and psychological outcomes. AR was underrepresented, with only one study. Risk of bias was generally low; however, publication bias could not be excluded due to limited funnel plot power (*n* < 10). **Conclusions:** VR, AR, and exergaming are effective adjuncts to conventional rehabilitation for CMDs, improving pain and function with high patient adherence. Nevertheless, gaps in long-term data, economic evaluation, and modality comparison persist. Future RCTs should address these limitations through standardized, inclusive, and longitudinal design.

## 1. Introduction

Chronic musculoskeletal disorders (CMDs), such as osteoarthritis, chronic low-back pain, fibromyalgia, shoulder dysfunction, and post-operative mobility impairments, constitute a major and escalating global health burden [1]. According to the Global Burden of Disease Study (2017), musculoskeletal conditions affect approximately 1.3 billion individuals worldwide, contributing to more than 138 million disability-adjusted life years (DALYs) and over 120,000 deaths annually. These disorders are characterized by persistent pain, stiffness, inflammation, and limited mobility lasting longer than three months, often resulting in significant functional decline, reduced quality of life, social isolation, and psychological comorbidities, including anxiety and depression [2]. Previous systematic reviews have examined the effects of virtual reality on various musculoskeletal conditions, but they often have a limited scope, present inconclusive findings, or contain substantial methodological weaknesses. However, none specifically focus on chronic cases with an integrated assessment of both functional and psychological outcomes. This review aims to address that gap with a rigorous synthesis of recent RCTs [3,4].

Chronic pain presents a particular challenge in rehabilitation, as it engages maladaptive neural mechanisms related to emotional distress and catastrophizing, which in turn reduce treatment adherence and slow functional recovery [1,3]. Conventional approaches—including physiotherapy, pharmacologic management, and lifestyle interventions—remain essential but are often constrained by poor long-term adherence, declining efficacy over time, and low motivation. Recurrence of symptoms and the psychosocial burden of chronic illness further limit the sustainability of traditional care strategies [3].

Emerging digital technologies have opened new avenues for rehabilitation, particularly through the application of virtual reality (VR) [5], augmented reality (AR) [6], and gamified exercise platforms, collectively referred to under the umbrella of extended reality (XR). XR encompasses interactive environments that integrate real and virtual stimuli to varying degrees, enhancing therapeutic engagement and outcome potential [5,7].

Virtual reality refers to fully immersive, computer-generated environments delivered via head-mounted displays that simulate functional tasks while simultaneously providing sensory distraction from pain [7]. Augmented reality, in contrast, overlays digital content onto the physical environment, typically via smart glasses, handheld devices, or tablet screens, allowing users to receive real-time visual feedback and corrective cues during movement [6]. Exergaming systems incorporate motion-tracking interfaces to link physical activity with gamified components, thereby promoting adherence, repetition, and motivation, particularly in populations with fibromyalgia, osteopenia, or joint dysfunction [8,9].

Given the variability in technological format, clinical focus, and therapeutic application, this review categorizes interventions into three core modalities: immersive VR, AR-based platforms, and non-immersive exergaming systems [1,2]. Examples include head-mounted immersive VR devices such as Oculus Rift, tablet-based AR rehabilitation programs, and Kinect- or Wii-based motion-driven gaming interfaces [10,11].

This systematic review aims to evaluate the effectiveness of VR, AR, and exergaming technologies in the rehabilitation of chronic musculoskeletal disorders in adults. The primary outcome is functional improvement, while secondary outcomes include pain reduction, enhancement of balance, and reduction in kinesiophobia, as assessed by validated outcome instruments such as the Western Ontario and McMaster Universities Osteoarthritis Index (WOMAC), Visual Analog Scale (VAS), EuroQol 5 Dimensions Questionnaire (EQ-5D), Tampa Scale for Kinesiophobia (TSK), and Short-Form 36 Health Survey (SF-36). As these technologies become increasingly accessible and clinically relevant, it is critical to assess their comparative efficacy, feasibility, and integration potential within contemporary rehabilitation practice.

Abbreviations: Augmented reality (AR), bone metabolism parameters (BMP), bone mineral density (BMD), Center for Epidemiologic Studies Depression Scale (CES-D), chronic musculoskeletal disorders (CMDs), continuous passive motion (CPM), disability-adjusted life years (DALYs), EuroQol 5 Dimensions—5 Levels (EQ-5D-5L), Fibromyalgia Impact Questionnaire (FIQ), Fracture Risk Assessment Tool (FRAX), hip fracture probability (HFP), health-related quality of life (HRQoL), Hospital for Special Surgery Knee Score (HSS), Knee injury and Osteoarthritis Outcome Score (KOOS), McGill Pain Questionnaire (MPQ), Modified Star Excursion Balance Test (mSEBT), numeric rating scale (NRS), osteoarthritis (OA), pressure pain threshold (PPT), Preferred Reporting Items for Systematic Reviews and Meta-Analyses (PRISMA), Patient-Reported Outcomes Measurement Information System (PROMIS), randomized controlled trial (RCT), Roland–Morris Disability Questionnaire (RMDQ), Scapular Assistance Test (SAT), Short-Form 36 Health Survey (SF-36), muscle oxygen saturation (SmO_2_), Shoulder Pain and Disability Index (SPADI), State–Trait Anxiety Inventory (STAI), Tampa Scale for Kinesiophobia (TSK), Timed Up-and-Go Test (TUG), visual analog scale (VAS), virtual reality (VR), Western Ontario and McMaster Universities Osteoarthritis Index (WOMAC), extended reality (XR).

## 2. Methods

This systematic review was conducted in accordance with the PRISMA 2020 (Preferred Reporting Items for Systematic Reviews and Meta-Analyses) guidelines [12]. The review protocol was registered in the International Prospective Register of Systematic Reviews [13] (PROSPERO; ID: CRD42024589007). As this study involved the synthesis of previously published data, ethical approval and informed consent were not applicable. This section outlines the methodology used to identify, select, and analyze randomized controlled trials (RCTs) evaluating virtual-reality and extended-reality interventions for chronic musculoskeletal disorders.

### 2.1. Eligibility Criteria

Eligible studies met the following criteria: randomized controlled trial (RCT) design; adult participants (≥18 years) with chronic musculoskeletal disorders (CMDs) of at least three months’ duration (e.g., fibromyalgia, osteoarthritis, chronic low-back pain, chronic shoulder dysfunction); implementation of virtual-reality (VR), augmented-reality (AR), or exergaming interventions as part of physical rehabilitation; the presence of a comparison group receiving standard physiotherapy, home-based exercise, or no treatment; and at least one outcome assessed with a validated instrument. 

Exclusion criteria included studies involving acute injuries, neurological or autoimmune conditions, non-RCT designs (e.g., protocols, case reports), non-English publications lacking validated translations, and unavailable full-text articles. When necessary, authors were contacted to retrieve missing outcome data.

### 2.2. Information Sources and Search Strategy

A comprehensive and systematic literature search was conducted across four electronic databases: PubMed, Scopus, PEDro, and the Cochrane Central Register of Controlled Trials (CENTRAL), covering all records through 1 May 2025. The search strategy was constructed based on the PICO framework and combined keywords and MeSH terms related to technology (e.g., “virtual reality”, “XR”, “exergaming”), clinical condition (e.g., “fibromyalgia”, “osteoarthritis”), and therapeutic context (e.g., “rehabilitation”, “physiotherapy”) using Boolean logic and truncation symbols. A pilot search was conducted to verify retrieval accuracy, and the full syntax for each database is detailed in Appendix A.

To minimize publication bias, additional searches were conducted in grey literature sources such as Google Scholar, OpenGrey, and ClinicalTrials.gov. Although the initial search applied no language restrictions, non-English studies were excluded during screening due to the unavailability of validated outcome measure translations. All records were screened using Rayyan QCRI by two independent reviewers; disagreements were resolved through discussion or consultation with a third reviewer. Study inclusion and exclusion procedures are outlined in the PRISMA 2020 flow diagram (Figure 1).

### 2.3. PICO Framework

The study selection criteria were explicitly structured using the PICO framework. The population comprised adults (≥18 years) diagnosed with chronic musculoskeletal conditions of at least three months’ duration, including post-surgical functional limitations. Interventions included extended reality (XR)-based rehabilitation technologies, classified into immersive VR systems (e.g., head-mounted displays), AR platforms (e.g., smart glasses, tablets), and non-immersive exergaming systems (e.g., Nintendo Wii, Microsoft Kinect), each applied with therapeutic objectives such as pain distraction, motor re-education, or functional training. Comparators involved conventional physiotherapy, structured home-based rehabilitation, lifestyle education, or waitlist control. Studies employing comparators deviating substantially from routine care were excluded to maintain consistency.

Outcome measures encompassed validated instruments assessing physical function (WOMAC, KOOS, Arm Curl Test, Timed Up-and-Go [TUG]), pain (VAS, NRS, MPQ), balance and mobility (mSEBT, TUG), psychological correlates of disability (TSK, CES-D, STAI), and health-related quality of life (EQ-5D, SF-36, and general QoL indices). When multiple outcomes were reported, primary post-intervention, time-matched comparisons were prioritized for synthesis.

### 2.4. Data Extraction and Management

Two reviewers (TP and PS) independently extracted study data using a standardized, pilot-tested Excel sheet. Extracted variables included study identifiers, population characteristics, intervention and control protocols, sample sizes, settings, duration, and results. Any discrepancies were resolved by consensus or by consulting a third reviewer (GG).

### 2.5. Risk of Bias Assessment

Methodological quality was assessed using the PEDro Scale (score range: 0–10) and the Downs and Black Checklist (maximum score: 28). The PEDro Scale was used to evaluate physiotherapy-specific internal validity, while the Downs and Black Checklist captured broader dimensions of bias, external validity, and reporting. Studies scoring ≥6 on PEDro or ≥20 on Downs and Black were considered good quality. Assessments were independently conducted by two reviewers, with disagreements resolved through discussion or third-party adjudication.

### 2.6. Effect Measures and Synthesis Methods

A quantitative synthesis was conducted where outcome data were sufficiently homogeneous across studies. For continuous outcomes, standardized mean differences (SMDs) with 95% confidence intervals (CIs) were calculated. Statistical heterogeneity was assessed using the I^2^ statistic, with values over 50% indicating moderate to high heterogeneity. Meta-analyses were performed using the DerSimonian–Laird random-effects model to account for clinical and methodological variability. All statistical analyses were conducted in SPSS (version 29) using the Meta-Analysis for Continuous Outcomes extension.

### 2.7. Assessment of Reporting Bias

Publication bias was explored through the visual inspection of funnel plots, where at least 10 studies were available per outcome. Additionally, the Egger’s regression test was applied to assess statistical asymmetry.

### 2.8. Certainty of Evidence

The certainty of evidence for each synthesized outcome was evaluated using the GRADE (Grading of Recommendations, Assessment, Development, and Evaluation) approach [14]. Domains considered included risk of bias, inconsistency, indirectness, imprecision, and publication bias. Each outcome was assigned a level of certainty: high, moderate, low, or very low. GRADE summary tables are presented in Appendix A.

## 3. Results/Findings

### 3.1. Search and Study Selection

A systematic search was conducted in the PubMed (*n* = 115), PEDro (*n* = 10), Cochrane Library (*n* = 48), and Scopus (*n* = 215) databases, yielding a total of 388 publications. After removing duplicates, 312 unique records remained for screening. Following title and abstract screening, 76 full-text articles were assessed for eligibility. Of these, 13 randomized controlled trials (RCTs) met the inclusion criteria and were included in the final qualitative synthesis, comprising a total of 881 participants aged 18 to 70 years.

### 3.2. Study Quality and Risk of Bias

The mean PEDro score was 6.31/10, reflecting overall good methodological quality. According to the PEDro scale, nine studies were rated as good quality (≥6) and four as fair quality, with the most common limitations involving lack of participant or therapist blinding, a structural limitation in behavioral and rehabilitation research.

In contrast, the average Downs and Black score was 22.8/28, indicating generally strong reporting and low risk of bias. Two studies were rated as excellent quality [15,16], nine as good [8,9,10,11,17,18,19,20,21], and two as fair [6,22]. Notably, studies focusing on fibromyalgia interventions, particularly using virtual reality (VR) or exergaming, achieved high Downs and Black scores (mean 22.8/28), despite scoring moderately on the PEDro scale due to an inherent lack of blinding [8,9,16,20,21,23].

This discrepancy underscores how PEDro may underestimate study rigor in trials where blinding is not feasible but reporting and internal validity remain strong. A summary of PEDro-based quality assessment for all included studies is presented in Table 1.

Therefore, employing both tools in parallel provided a more comprehensive appraisal of trial quality and bias risk. The combined PEDro and Downs and Black quality scores are detailed in Table 2.

### 3.3. Geographical Distribution

The included studies were conducted across eight countries. Spain led with five studies, followed by the USA and Turkey with two each. Other contributing countries included China, Iran, India, Pakistan, and Germany. Most fibromyalgia studies were conducted in Spain, leveraging culturally adapted interventions like VirtualEx-FM [20].

### 3.4. VR/AR Technologies Used

The reviewed interventions were grouped into four main technological categories based on the degree of immersion, interactivity, and delivery format. Fully immersive virtual-reality (VR) systems (*n* = 4), such as those using Oculus Rift or HTC Vive headsets, provide sensory-rich simulations designed to facilitate motor learning and cognitive engagement. Exergaming platforms (*n* = 5), including the Kinect, Nintendo Wii, and VirtualEx-FM systems, offer non-immersive, game-based rehabilitation experiences that emphasize movement repetition and motivation. Augmented-reality (AR) interventions (*n* = 1), such as the protocol described in [6], overlay digital guidance onto real-world tasks to enhance accuracy and feedback in post-operative rehabilitation. Finally, motion-tracking and tele-rehabilitation systems (*n* = 3), including remote Kinect-based training programs, support supervised home-based therapy without immersive visual environments.

Among these modalities, exergaming yielded the most consistent improvements in physical function and quality of life across diverse clinical populations. In contrast, immersive VR demonstrated stronger effects on psychological outcomes such as depression and kinesiophobia, particularly in populations with fibromyalgia and chronic low-back pain (e.g., [10,21,23]).

### 3.5. Adherence and Response Rates

Adherence ranged from 81% to 93%, with the highest rates observed in long-term, gamified interventions like the 24-week exergame programs [20]. Most studies demonstrated improvements in multiple outcomes, with pain, physical function, and quality of life being the most consistently enhanced variables.

### 3.6. Study Characteristics

The 13 randomized controlled trials (RCTs) included in this review incorporated a variety of extended reality technologies aimed at musculoskeletal rehabilitation. To facilitate comparative interpretation, the interventions were grouped into four primary modalities based on technological features and therapeutic application. These included immersive virtual-reality systems delivering sensorimotor simulations through head-mounted displays, augmented-reality platforms providing real-time visual cues during physical tasks, non-immersive exergaming interfaces such as Kinect and Wii that emphasized repetitive functional movement, and motion-tracking tools designed for remote physiotherapy without immersive environments. This framework reflects the technological diversity within the field and supports a structured synthesis of outcomes across trials. The selected studies, drawn from an initial pool of 388 records, encompassed 881 adults aged 18 to 70 years and addressed a wide range of conditions—including orthopedic, rheumatologic, and neurologic disorders—underscoring the clinical versatility of digital rehabilitation.

#### 3.6.1. Demographics and Gender Distribution

Gender representation across the included studies demonstrated notable variability. Five studies focused exclusively on women with fibromyalgia, comprising a total of 281 participants (studies 7, 8, 14, 18, and 19). This reflects the higher prevalence of fibromyalgia in females and a research focus on digital rehabilitation within this specific population. In contrast, one study targeted only male participants, specifically young adult military recruits (study 17).

The remaining seven studies included mixed-gender cohorts, although women consistently represented the majority, accounting for at least 60–65% of total participants. Across these studies, women showed notable responsiveness to virtual-reality interventions, particularly within exergaming contexts, reporting significant improvements in pain levels, physical function, quality of life, and balance.

This gender imbalance introduces potential limitations in the generalizability of findings, particularly regarding underrepresented populations such as older males. It also highlights a broader methodological gap in digital rehabilitation research and emphasizes the need for more inclusive and demographically balanced recruitment strategies in future clinical trials to enhance representativeness and equitable applicability of outcomes.

#### 3.6.2. Target Conditions

The included studies encompassed a broad spectrum of chronic musculoskeletal pathologies, reflecting the clinical versatility of VR/AR-based rehabilitation interventions. Knee-related conditions represented the most frequently studied category, comprising approximately 30.7% of trials (e.g., [6,17,19]). Fibromyalgia accounted for 23% of the study population, with interventions tailored to this condition evaluated in multiple trials [8,9,16,20]. An additional 15.3% of studies focused on shoulder or upper-limb dysfunction, often in the context of subacromial impingement or scapular dyskinesis [16,22]. Similarly, 15.3% of studies addressed lower-back or patellofemoral pain syndromes [10,15], while the remaining 15.3% examined bone health-related conditions, specifically osteopenia and osteoporosis [11]. Τhis distribution illustrates the growing interest in applying digital rehabilitation tools across a variety of chronic pain and mobility-limiting conditions, with fibromyalgia and degenerative joint disorders emerging as key focus areas.

#### 3.6.3. Intervention Design and Delivery

The duration of the intervention programs varied substantially across studies, ranging from as short as 3 days to as long as 24 weeks, reflecting the heterogeneity in rehabilitation protocols. Approximately 75% of the trials employed supervised delivery formats—either in-person or through tele-rehabilitation—emphasizing the importance of guided therapeutic engagement. Several studies implemented home-based or hybrid models, particularly those in [16,17], which aimed to enhance accessibility and adherence. The technological complexity of the interventions also varied, from basic motion-tracking platforms to advanced, multimodal VR systems incorporating neuromodulatory components, as seen in [9]. This range highlights both the adaptability and scalability of digital rehabilitation strategies in diverse clinical settings.

#### 3.6.4. Comparators and Methodological Notes

A comprehensive summary of all intervention protocols, control groups, and main outcomes is provided in Table 3 below. Across the included trials, control groups consistently adhered to conventional rehabilitation protocols, including standard physiotherapy, educational interventions, or no treatment conditions, thereby providing an appropriate baseline for comparative evaluation. Although the study designs shared structural similarities, substantial variability was observed in intervention formats, target conditions, outcome measures, and durations. This degree of clinical and methodological heterogeneity limited the feasibility of direct comparison among studies across all outcome domains. However, a meta-analysis was conducted for pain outcomes, where sufficient data harmonization was possible. Therefore, a narrative synthesis was employed for the remaining outcomes, a decision that aligns with prior systematic reviews in the field [4], which have similarly encountered substantial inter-study variability in digital rehabilitation research.

### 3.7. Meta-Analysis and Narrative Synthesis Justification

#### 3.7.1. Exploratory Meta-Analysis of Pain Outcomes

To complement the narrative synthesis, an exploratory meta-analysis was conducted to evaluate the impact of virtual- and augmented-reality (VR/AR) interventions on pain intensity, using the visual analog scale (VAS) as the primary outcome. Eight randomized controlled trials (RCTs) provided sufficient statistical data—including post-intervention means and standard deviations—from which standardized mean differences (SMDs) and standard errors were derived using established conversion formulas.

A random-effects model was applied using the DerSimonian–Laird method in SPSS version 29, employing the Meta-Analysis for Continuous Outcomes extension. The pooled effect size was −1.14 (95% CI: −1.63 to −0.75), indicating a large and statistically significant reduction in pain favoring immersive and digitally mediated interventions. No statistical heterogeneity was observed (Q = 2.73; I^2^ = 0%; τ^2^ = 0.00), suggesting a consistent effect across trials despite clinical diversity. However, the assumptions required for a robust meta-analysis were only partially met due to notable heterogeneity in both intervention protocols and outcome measures across studies.

These findings corroborate prior evidence supporting the analgesic efficacy of immersive technologies in musculoskeletal rehabilitation, especially in conditions such as fibromyalgia, osteoarthritis, and post-operative recovery. However, the small number of included studies (*n* = 8) precluded formal funnel plot asymmetry testing, in accordance with Cochrane guidelines. Despite this, the uniform directionality of effect estimates suggests a potential risk of publication bias, and pooled estimates should be interpreted with appropriate caution.

Given the small number of eligible studies and the heterogeneity in interventions, populations, and outcome instruments, the current meta-analysis is characterized as exploratory. This designation reflects the preliminary and hypothesis-generating nature of the synthesis, intended to estimate effect sizes while acknowledging underlying variability. Similar approaches have been adopted in the literature when full meta-analytic assumptions (e.g., outcome homogeneity, intervention consistency) are only partially met. As such, the results should be interpreted as indicative rather than confirmatory, warranting future validation through high-quality trials with greater standardization (Cochrane Handbook, Ch. 10.10.1).

#### 3.7.2. Narrative Results and Clinical Variability

Due to substantial heterogeneity among the included RCTs, meta-analysis was not feasible for secondary outcomes, including functional status, psychological parameters, and postural control. Instead, a structured narrative synthesis was employed, organized by intervention type and outcome domain.

Technology Variability: Interventions varied considerably, encompassing immersive VR (e.g., Pro-Kin, VR Dodgeball), non-immersive exergaming platforms (e.g., Nintendo Wii, Microsoft Kinect), augmented-reality overlays, and hybrid neuromodulatory systems such as EXOPULSE. Comparator arms ranged from conventional physiotherapy to home exercise, lifestyle education, or waitlist control, introducing further methodological inconsistency.

Clinical Diversity: Participant cohorts spanned multiple chronic musculoskeletal conditions, including fibromyalgia, knee osteoarthritis, patellofemoral pain, shoulder dysfunction, and osteopenia. Variability in disease etiology and baseline function limited the comparability of rehabilitation responses across studies.

Outcome Measurement Inconsistency: A wide range of validated tools were used across domains—VAS for pain, WOMAC and KOOS for physical function, SF-36 and EQ-5D-5L for quality of life, CES-D for depression, and TSK for fear-avoidance—complicating unified interpretation due to differences in psychometric properties and sensitivity.

Functional Outcomes: Clinically meaningful gains were observed in mobility, strength, and flexibility across several trials. For example, sustained improvements were reported over 24 weeks of exergaming in women with fibromyalgia [8], while enhanced joint function and range of motion were demonstrated in patellofemoral and shoulder pathologies in [15] and [18], respectively. This suggests more consistent effectiveness in short-term functional outcomes compared to long-term effects.

Psychological Outcomes: Some studies suggested beneficial effects on anxiety, mood, and motivation, particularly in elderly populations undergoing orthopedic rehabilitation [16,18]. However, only a minority employed standardized psychological measures, such as the CES-D or the mental component of the SF-36, thereby limiting definitive conclusions regarding emotional or cognitive outcomes. In conclusion, our findings indicate that virtual-reality interventions offer promising short-term benefits. However, further large-scale RCTs are needed to confirm long-term effects.

Balance and Safety Considerations: Immersive VR interventions incorporating proprioceptive feedback (e.g., Pro-Kin) were associated with improved postural control and reduced kinesiophobia in cohorts with chronic low-back pain or osteopenia [10,19]. Safety monitoring, however, was inconsistently reported. Only three studies explicitly addressed adverse effects such as cybersickness, dizziness, or fatigue [9,10,18], highlighting the need for more systematic risk assessment, particularly for balance-impaired or older individuals.

#### 3.7.3. Certainty of Evidence (GRADE Assessment)

To complement the quantitative synthesis, the certainty of evidence for the primary outcome—pain reduction—was evaluated using the GRADE (21) (Grading of Recommendations Assessment, Development, and Evaluation) approach. The overall level of certainty was rated as moderate. This judgment reflects the generally low to moderate risk of bias across the included studies, as assessed through both the PEDro scale (mean score 6.31/10) and the Downs and Black checklist (mean score 22.8/28). Although participant and therapist blinding was rarely feasible, internal validity was preserved across trials. No statistical heterogeneity was observed (I^2^ = 0%), indicating consistency of effect estimates. The population, interventions, and outcomes directly aligned with the review objective, and thus, no serious concerns of indirectness were identified. However, imprecision was considered moderate due to the relatively small number of contributing studies (*n* = 8 RCTs) and the presence of wide confidence intervals in some cases. Furthermore, the potential for publication bias cannot be excluded, given the consistent direction of positive results and the inability to formally assess funnel plot asymmetry with fewer than ten studies. Overall, the current body of evidence provides moderately strong support for the analgesic effectiveness of virtual- and augmented-reality interventions in chronic musculoskeletal rehabilitation. A GRADE evidence summary is provided in Appendix A.

## 4. Discussion

This discussion synthesizes evidence from 13 randomized controlled trials to evaluate the effectiveness, variability, and limitations of virtual and augmented reality-based rehabilitation strategies for chronic musculoskeletal disorders. It contextualizes findings within clinical, technological, and methodological frameworks to guide future research and practice.

The cross-analysis of gender, geography, and technology suggests that women with chronic musculoskeletal conditions—particularly those with fibromyalgia—may derive notable benefits from non-immersive exergaming programs, likely due to their high levels of engagement and the functional improvements reported in trials [8,16]. This trend underscores the importance of tailoring interventions to patient-specific needs, including gender-sensitive program design.

Furthermore, the implementation of VR and AR technologies across diverse geographic settings, from Europe to Asia and North America, indicates a high degree of adaptability and translational potential in different healthcare systems [17]. These findings support the broader clinical argument for personalized, technology-enabled rehabilitation strategies that integrate patient characteristics with context-specific delivery models.

This systematic review examined the effectiveness of virtual-reality (VR) and augmented-reality (AR) interventions in the management of chronic musculoskeletal disorders (CMDs). Based on 13 randomized controlled trials involving a total of 881 participants, the reviewed technologies demonstrated substantial potential in improving pain, physical function, balance, and health-related quality of life (HRQoL) across a range of musculoskeletal and rheumatic conditions.

Pain reduction was the most frequently reported outcome, typically evaluated using the visual analog scale (VAS) or condition-specific tools such as the Western Ontario and McMaster Universities Osteoarthritis Index (WOMAC) and the Fibromyalgia Impact Questionnaire (FIQ). Significant reductions in pain were observed in several studies employing VR or AR interventions, including [6,8,19]. Non-immersive exergaming programs, particularly VirtualEx-FM, also demonstrated meaningful reductions in perceived pain and health burden, as shown in two fibromyalgia trials [8,20]. Notably, promising outcomes through the integration of wearable neuromodulation with VR were reported, resulting in both pain reduction and improved tissue oxygenation [9].

Functional outcomes, including disability and mobility, were improved across various domains such as range of motion, balance, and activity-specific performance. Improvements were reported in individuals with patellofemoral pain [15], patients with shoulder dysfunction [18], and those with chronic low-back pain following VR-based task training [20]. Long-term functional improvements were also demonstrated in strength, flexibility, and mobility after a 24-week exergaming intervention in women with fibromyalgia [20].

While AR-based interventions were less represented in the evidence base compared to VR and exergaming, one study illustrated that AR-guided rehabilitation can facilitate accelerated recovery and pain relief following knee surgery, underscoring the emerging relevance of AR in post-operative care [6].

Psychological outcomes such as reduced anxiety and lower perceived exertion were noted in studies involving older adults undergoing orthopedic rehabilitation [16,18]. Nevertheless, only a limited number of trials formally assessed mental-health parameters using validated instruments, indicating a notable gap in understanding the broader biopsychosocial effects of VR/AR-based interventions.

### 4.1. Identified Gaps in the Literature

Despite promising findings, this review highlights several unresolved issues in the current evidence base for VR, AR, and exergaming in chronic musculoskeletal rehabilitation. One major gap is the lack of long-term follow-up data. None of the included studies reported outcomes beyond six months, leaving the durability of observed improvements uncertain, particularly in chronic and relapsing conditions such as fibromyalgia and osteoarthritis [8,20]. Another critical issue is the absence of direct comparative effectiveness trials. No study directly compared VR to AR or immersive to non-immersive systems, limiting any conclusions regarding the relative superiority of digital modalities. The only identified AR intervention was included in a single trial [6].

In addition, the current literature lacks formal economic evaluations. Although some studies suggested potential reductions in healthcare resource use and hospital readmissions [6,17], none implemented structured cost-benefit or cost-effectiveness analyses, thereby hindering assessments of financial sustainability or reimbursement feasibility. Sample diversity also presents a limitation, as many trials included gender-restricted or age-limited populations—such as female-only cohorts [8,11] or young male participants [19]—which restricts the generalizability of findings to broader clinical settings.

Safety outcomes were inconsistently tracked. Only a minority of trials explicitly monitored adverse events. One study examined feasibility and safety in VR-based training for chronic low-back pain, highlighting cybersickness and fall risk as important considerations in immersive settings [20]. Another incorporated physiological markers, such as muscle oxygen saturation (SmO_2_), as proxies for systemic safety and exertion levels in patients with fibromyalgia [9]. Reductions in anxiety levels were reported, although adverse events were not systematically monitored [7]. These omissions are especially relevant for vulnerable populations such as older adults or individuals with balance impairments. Furthermore, while some interventions suggested psychological or motivational benefits [7,16], few studies employed validated psychosocial scales. As a result, the potential emotional and cognitive impact of immersive rehabilitation technologies remains poorly understood.

Finally, no study addressed ethical or regulatory issues such as data privacy, digital consent procedures, or implementation frameworks—key areas that must be considered in future clinical applications of VR and AR technologies.

### 4.2. Limitations of This Systematic Review

This systematic review presents several methodological limitations that should be considered when interpreting its findings. Although an exploratory meta-analysis on pain outcomes was successfully conducted and yielded a significant pooled effect, substantial heterogeneity in intervention modalities, outcome measures, follow-up durations, and technology types (e.g., immersive VR versus non-immersive exergaming) precluded a full meta-analysis across all outcome domains. As such, narrative synthesis was adopted as the primary analytical approach for secondary outcomes.

While the included studies were generally of fair to good quality—as reflected in a mean PEDro score of 6.31/10 and a Downs and Black average of 22.8/28—only two trials achieved an “excellent” methodological rating. Participant and therapist blinding was rarely feasible, further limiting internal validity. The review also excluded grey literature (e.g., conference proceedings, dissertations) and non-English publications, increasing the potential for publication bias and possibly omitting relevant research from non-English-speaking countries.

Although two validated tools were used for bias assessment, the Cochrane Risk of Bias 2.0 (RoB 2.0) tool was not applied, which limits methodological alignment with Cochrane-compliant systematic reviews. Furthermore, several technologies evaluated in the included trials—such as Wii Fit and early-generation Kinect—are becoming obsolete, thereby reducing the applicability of findings to modern digital rehabilitation tools like Oculus Quest, EXOPULSE, or AI-assisted systems.

Lastly, adverse effects such as cybersickness, fatigue, or dizziness were inconsistently reported. The absence of systematic safety monitoring across the primary studies restricted this review’s ability to assess risk profiles accurately—an issue of particular concern for older adults and neurologically vulnerable populations. These limitations emphasize the need for future research to adopt standardized, longitudinal, and technologically current protocols in order to establish the safety, efficacy, and implementation readiness of VR and AR interventions for musculoskeletal rehabilitation.

### 4.3. Implications for Physiotherapy Practice

The integration of virtual reality (VR), augmented reality (AR), and exergaming into physiotherapy practice offers promising avenues to enhance rehabilitation outcomes for patients with chronic musculoskeletal disorders. Across the reviewed trials, these digital interventions improved patient motivation, engagement, and adherence, particularly in longer-term programs such as the 24-week exergaming protocols for fibromyalgia [8,20]. Immersive VR applications, including task-based environments like dodgeball training [10] and Pro-Kin balance platforms [19], facilitated neuromuscular retraining and reduced kinesiophobia. Tele-rehabilitation systems employing motion-tracking demonstrated non-inferiority to in-clinic care [16,17], while AR protocols supported post-surgical recovery and precision training [6].

To enable clinical implementation, future practice should prioritize training physiotherapists in sensor interpretation and digital session management, developing evidence-informed clinical guidelines for VR/AR protocols, and establishing reimbursement models that reflect technological resource use. Emerging tools that merge VR with neuromodulation or artificial intelligence—such as the EXOPULSE system—may further expand physiotherapy capabilities, particularly in remote or hybrid care models that support personalized, accessible rehabilitation.

## 5. Conclusions

This systematic review supports the clinical potential of virtual reality (VR), augmented reality (AR), and exergaming as complementary tools in the rehabilitation of chronic musculoskeletal disorders. These technologies have demonstrated benefits in improving pain, mobility, and patient engagement across various conditions.

However, variability in study design and outcome reporting, along with limited long-term and economic data, prevents the development of definitive clinical guidelines. Future research should address these gaps to ensure evidence-based integration of digital interventions into routine physiotherapy practice.

## Figures and Tables

**Figure 1 bioengineering-12-00745-f001:**
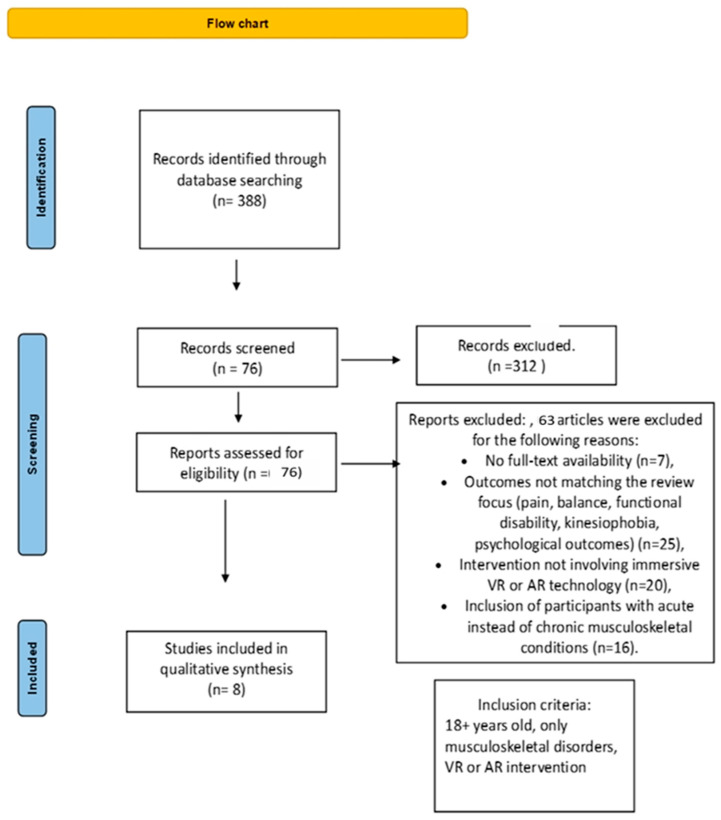
PRISMA 2020 flow diagram of study selection process.

**Table 1 bioengineering-12-00745-t001:** PEDro scale—quality assessment of included studies.

Study	1	2	3	4	5	6	7	8	9	10	11	Score
Bettger et al. (2020) [23]	+	+	−	+	−	−	−	+	+	+	+	6/10
Fuchs et al. (2022) [18]	+	+	−	+	−	−	−	−	−	+	+	4/10
Pekyavas et al. (2017) [22]	+	+	+	+	−	−	−	+	−	+	+	7/10
Ebrahimi et al. (2021) [15]	+	+	+	+	−	−	+	+	+	+	−	7/10
Thomas et al. (2016) [10]	−	+	+	+	−	−	−	+	−	+	+	6/10
Nambi et al. (2020) [19]	+	+	+	+	+	−	+	+	+	+	+	9/10
Li et al. (2022) [6]	+	+	−	+	−	−	−	+	+	+	+	6/10
Riaz et al. (2024) [11]	+	+	+	+	+	−	+	+	+	+	+	9/10
Villafaina et al. (2019) [8]	+	+	−	+	−	−	+	+	+	+	+	7/10
Martín-Martínez et al. (2019) [20]	+	+	−	+	−	−	+	+	+	+	+	7/10
Garcia-Palacios et al. (2015) [16]	+	+	−	+	−	−	−	+	−	+	+	5/10
Gulsen et al. (2022) [21]	+	+	−	+	−	−	+	−	−	+	+	5/10
Rubio-Zarapuz et al. (2024) [9]	−	+	−	+	−	−	−	−	−	+	+	4/10

**Average PEDro quality score:** 6.31/10—indicating an overall good methodological quality of the included studies. Note: according to PEDro guidelines, Item 1 (eligibility criteria) is not included in the final score. Scores range from 0–10 based on Items 2–11. “+” indicates that the criterion is fulfilled. “−” indicates that the criterion is not fulfilled.

**Table 2 bioengineering-12-00745-t002:** Downs and Black quality assessment of included RCTs.

RCT	PEDro Score	Downs and Black Score	Quality of Study
Bettger et al. (2020) [23]	6/10	25/28	Good
Fuchs et al. (2022) [18]	4/10	22/28	Good
Pekyavas et al. (2017) [22]	7/10	17/28	Fair
Ebrahimi et al. (2021) [15]	7/10	27/28	Excellent
Thomas et al. (2016) [10]	6/10	22/28	Good
Nambi et al. (2020) [19]	9/10	24/28	Good
Li et al. (2022) [6]	6/10	16/28	Fair
Riaz et al. (2024) [11]	9/10	23/28	Good
Villafaina et al. (2019) [8]	7/10	24/28	Good
Martín-Martínez et al. (2019) [20]	7/10	25/28	Good
Garcia-Palacios et al. (2015) [16]	5/10	26/28	Excellent
Gulsen et al. (2022) [21]	5/10	24/28	Good
Rubio-Zarapuz et al. (2024) [9]	4/10	22/28	Good

**Table 3 bioengineering-12-00745-t003:** Summary of randomized controlled trials (RCTs) on VR, AR, and exergaming interventions in chronic musculoskeletal disorders.

Study	Sample Size (*n*, Age)	Pathology	Intervention	Control Group	Outcome Measures	Main Results	Technology Type
Nambi et al. [19]	60 (18–25 y)	Post-traumatic OA	VR Pro-Kin System	Conventional physiotherapy	VAS, WOMAC, BMP, inflammatory markers	Improved pain, disability, inflammation (*p* < 0.001)	VR (immersive)
Pekyavas et al. [22]	30 (~41 y)	Shoulder dysfunction	Wii-based exergames	Home exercises	VAS, SPADI, Neer, Hawkins, SAT	Functional improvements (*p* < 0.05)	Exergaming
Li et al. [6]	40 (~33 y)	Post-op knee rehab	AR-based rehab program	Conservative therapy	HSS, VAS	Faster recovery, better HSS scores (*p* < 0.05)	AR (immersive)
Riaz et al. [11]	52 (48–70 y)	Osteopenia	Kinect-based VR training	Regular activity	BMD, FRAX, HFP	Improved BMD, fracture risk reduction (*p* < 0.001)	Exergaming
Thomas et al. [10]	52 (18–50 y)	Chronic low-back pain	VR dodgeball training	No intervention	TSK, CES-D, RMDQ, MPQ	Improved pain expectations (*p* = 0.001)	VR (immersive)
Ebrahimi et al. [15]	26 (~30 y)	Patellofemoral pain	Kinect VR therapy	Lifestyle education	VAS, Kujala, SF-36, mSEBT	Improved function and quality of life	VR (immersive)
Bettger et al. [23]	306 (~65 y)	TKA	Virtual PT (tele-rehab)	Traditional PT	KOOS, PROMIS	Non-inferior knee function, fewer readmissions (*p* = 0.007)	Motion-tracking
Fuchs et al. [18]	55 (~70 y)	TKA	VR-assisted CPM sessions	Conventional physiotherapy	VAS, WOMAC, STAI	Pain and anxiety reduction, no functional difference (*p* > 0.05)	VR (immersive)
Villafaina et al. [8]	55 (~54 y)	Fibromyalgia	24-week exergames (VirtualEx-FM)	No exercise	VAS, EQ-5D-5L	Improved HRQoL and pain (*p* < 0.05)	Exergaming
Martín-Martínez et al. [20]	55 (~54 y)	Fibromyalgia	24-week exergames (VirtualEx-FM)	No exercise	Arm curl, sit and reach, TUG	Improved strength, mobility, flexibility (*p* < 0.05)	Exergaming
Garcia-Palacios et al. [16]	61 (23–70 y)	Fibromyalgia	VR + activity management	Usual treatment	FIQ, QoL, Coping	Improved disability and quality of life	VR (non-immersive)
Rubio-Zarapuz et al. [9]	89 (18–67 y)	Fibromyalgia	EXOPULSE suit + VR	Neuromodulation/exercise/control	NRS, PPT, up-and-go, SmO_2_	Improvements in pain and oxygenation (*p* < 0.05)	VR + neuromodulation
Gulsen et al. [21]	20 (~45 y)	Fibromyalgia	Immersive VR + exercise (pilates + aerobic)	Exercise only (pilates + aerobic)	VAS, PPT, functional tests	Greater pain reduction and functional improvement	VR (immersive)

**Abbreviations:** VAS: visual analog scale; WOMAC: Western Ontario and McMaster Universities Osteoarthritis Index; BMP: bone metabolism parameters; SPADI: Shoulder Pain and Disability Index; HSS: Hospital for Special Surgery Score; BMD: bone mineral density; FRAX: Fracture Risk Assessment Tool; HFP: hip fracture probability; KOOS: Knee injury and Osteoarthritis Outcome Score; PROMIS: Patient-Reported Outcomes Measurement Information System; TSK: Tampa Scale for Kinesiophobia; CES-D: Center for Epidemiologic Studies Depression Scale; RMDQ: Roland–Morris Disability Questionnaire; MPQ: McGill Pain Questionnaire; Kujala: Kujala Anterior Knee Pain Scale; SF-36: Short-Form Health Survey; mSEBT: Modified Star Excursion Balance Test; STAI: State–Trait Anxiety Inventory; EQ-5D-5L: EuroQol 5 Dimensions—5 Levels; TUG: Timed Up-and-Go Test; FIQ: Fibromyalgia Impact Questionnaire; QoL: quality of life; NRS: numeric rating scale; PPT: pressure pain threshold; SmO_2_: muscle oxygen saturation; VAS: visual analog scale; SPADI: Shoulder Pain and Disability Index; Neer: Neer impingement test; Hawkins: Hawkins–Kennedy test; SAT: Scapular Assistance Test (Neer/Hawkins/SAT: orthopedic shoulder tests); HSS: Hospital for Special Surgery Knee Score.

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
