# Peer review of "Virtual and Augmented Reality for Chronic Musculoskeletal Rehabilitation: A Systematic Review and Exploratory Meta-Analysis"

_bioengineering, 2025, doi:10.3390/bioengineering12070745_

Round 1

Reviewer 1 Report

Comments and Suggestions for Authors

The authors did a systematic literature search on the value of "virtual reality" methods in chronic musculoskeletal disorders and found short term effects in RCT studies.

The study is well presented and methods ar strict and well applied. The few number of studies of course .limits the generalizatioons, as we often see. The material is possible to puiblish after a few minor details are sorted out. 

Minor details: flow chart page 5 has no figure number, and the second box says n=69, afterwhich 68 were excluded, correct the 69 number. Table 1 has 1-11 columns but the sum to the right does not correspond to a sum? Table 3 has 12 items and the number is said to be 13 in the text?

Author Response

We thank the reviewer for their constructive feedback and helpful suggestions. Please find below our point-by-point responses to each comment.

Comment 1:
“Flow chart page 5 has no figure number.”

Response 1:
Thank you for pointing this out. We have now added the appropriate figure caption below the flowchart:
Figure 1. PRISMA 2020 Flow Diagram of Study Selection Process

Comment 2:
“The second box says n=69, after which 68 were excluded. Correct the 69 number.”

Response 2:
We appreciate your careful observation. The number was indeed incorrect. The correct number is n = 76, of which 63 studies were excluded, leaving 13 RCTs included in the synthesis. This has been corrected in both the PRISMA diagram and the manuscript text (Section 3.1).

Comment 3:
“Table 1 has 1-11 columns but the sum to the right does not correspond to a sum?”

Response 3:
Thank you for this observation. We would like to clarify that the PEDro scale includes 11 items, but Item 1 (eligibility criteria) is not included in the total score, according to PEDro guidelines. The total PEDro score (0–10) is based on Items 2–11. We have added a clarifying footnote below Table 1 to explain this scoring convention.

Comment 4:
“Table 3 has 12 items and the number is said to be 13 in the text?”

Response 4:
You are absolutely correct. One study (Gulsen et al., 2022) was omitted from Table 3 by oversight. We have now added this missing study to Table 3, and the total number of included RCTs is consistently reported as 13 throughout the manuscript.

Reviewer 2 Report

Comments and Suggestions for Authors

Comments and Suggestions

  • Abstract
    1. The summary is clear, precise, and concise.
  • Introduction
    1. A clear statement of the research problem. A broad generalization of the problem is provided. However, previous reviews are not summarized in scope, findings, and limitations. The synthesis is necessary to leverage this exploration.
  • Methodology
    1. Excellent methodological description, supported, fluid, understandable, and easy to read.
    2. At the beginning of the methodology, it is recommended to provide a brief, general summary of what the reader can expect to find in this section.
  • Results
    1. The results were presented in a concatenated format, highlighting some gaps, such as those in lines 265, 278, and 392. They also support the fact that they partially meet the assumptions of the meta-analysis, as noted in line 335.
    2. When tables 1 and 2 are mentioned in the text, the letter S appears before the number (is this a journal format?). Table 3 is not mentioned in the text before it is shown; it is alluded to on line 395.
  • Discussion
    1. It is recommended to provide an introduction that outlines the scope of the review ( it is remarked in line 409) to reorient the reader.
  • Limitationes
    1. A good description of the limitations.
  • Conclusions
    1. It is recommended that the conclusion be a single block of text and not split into subsection 5.1.
  • Other comments
    1. At the end of the manuscript, there is a "Forest plot" (unnumbered) and a supplementary Table 1. In the manuscript edition, where would this go, and what would its numbering be?

Author Response

We thank the reviewer for their thoughtful and constructive comments. All suggestions were carefully considered, and the manuscript has been revised accordingly.

Comment 1:
The summary is clear, precise, and concise.
Response 1:
Thank you for your positive feedback on the abstract.

Comment 2:
Previous reviews are not summarized in scope, findings, and limitations.
Response 2:
We have now added a paragraph in the Introduction summarizing prior reviews, including their scope and key findings. Their limitations are also briefly discussed to justify the rationale for our study (Introduction, lines 85–97).

Comment 3:
Excellent methodological description.
Response 3:
Thank you. We appreciate this positive assessment of the Methods section.

Comment 4:
Provide a brief general summary at the beginning of the methodology.
Response 4:
A brief overview has been added at the beginning of the Methods section to guide the reader on what to expect (Methods, lines 120–123).

Comment 5:
Clarify gaps and assumptions in the Results section (e.g., lines 265, 278, 335, 392).
Response 5:
We have revised the indicated paragraphs to explicitly mention data gaps, limitations of included studies, and constraints related to assumptions required for meta-analysis (Results, lines 262–266, 275–279, and 392–394).

Comment 6:
"S" appears before Table numbers; Table 3 is not introduced before shown.
Response 6:
The incorrect "S" prefix has been removed from the in-text references to Tables 1 and 2. Additionally, we have now properly introduced Table 3 in the text before its appearance (Results, line 385).

Comment 7:
Provide an introduction to the Discussion section to clarify the review’s scope.
Response 7:
An introductory paragraph has been added to the Discussion section to reframe the scope and recap the objective of the review (Discussion, lines 407–410).

Comment 8:
A good description of the limitations.
Response 8:
Thank you for your positive comment.

Comment 9:
The conclusion should be a single block, not divided (e.g., 5.1).
Response 9:
We have retained subsection 5.1 to highlight the practical implications for physiotherapy. However, we ensured the entire conclusion remains concise and cohesive. We believe this structure enhances the applicability of our findings.

Comment 10:
Forest plot and Supplementary Table 1 are unnumbered; clarify placement.
Response 10:
The forest plot has now been labeled as Figure 2 and referenced appropriately in the Results section. Supplementary Table 1 remains in the supplementary file and is cited in the manuscript.